# PCLast: Discovering Plannable Continuous Latent States

## Abstract

Goal-conditioned planning benefits from learned low-dimensional representations of rich, high-dimensional observations. While compact latent representations, typically learned from variational autoencoders or inverse dynamics, enable goal-conditioned planning they ignore state affordances, thus hampering their sample-efficient planning capabilities. In this paper, we learn a representation that associates reachable states together for effective onward planning. We first learn a latent representation with multi-step inverse dynamics (to remove distracting information); and then transform this representation to associate reachable states together in $\ell_2$ space. Our proposals are rigorously tested in various simulation testbeds. Numerical results in reward-based and reward-free settings show significant improvements in sampling efficiency, and yields layered state abstractions that enable computationally efficient hierarchical planning.

## 1 Introduction

Deep reinforcement learning (RL) has emerged as a choice tool in mapping rich and complex perceptual information to compact low-dimensional representations for onward (motor) control in virtual environments Silver et al. (2016), software simulations Brockman et al. (2016), and hardware-in-the-loop tests Finn & Levine (2017). Its impact traverses diverse disciplines spanning games (Moravčík et al., 2017; Brown & Sandholm, 2018), virtual control (Tunyasuvunakool et al., 2020), healthcare (Johnson et al., 2016), and autonomous driving (Maddern et al., 2017; Yu et al., 2018). Fundamental catalysts that have spurred these advancements include progress in algorithmic innovations (Mnih et al., 2013; Schrittwieser et al., 2020; Hessel et al., 2017) and learned (compact) latent representations (Bellemare et al., 2019; Lyle et al., 2021; Lan et al., 2022; Rueckert et al., 2023; Lan et al., 2023).

Latent representations, typically learned by variational autoencoders (Kingma & Welling, 2013) or inverse dynamics (Paster et al., 2020; Wu et al., 2023), are mappings from high-dimensional observation spaces to a reduced space of essential information where extraneous perceptual information has already been discarded. These compact representations foster sample efficiency in learning-based control settings (Ha & Schmidhuber, 2018; Lamb et al., 2022). Latent representations however often fail to correctly model the underlying states' affordances. Consider an agent in the 2D maze of Fig. 1a. A learned representation correctly identifies the agent's (low-level) position information; however, it ignores the scene geometry such as the wall barriers so that states naturally demarcated

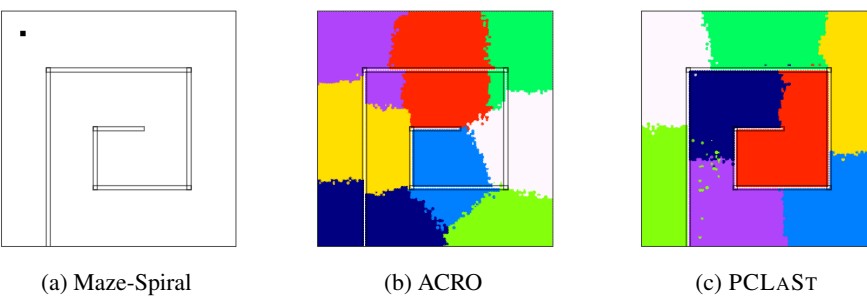

|  (a) Maze-Spiral | (b) ACRO | (c) PCLaSt |
|---|---|---|

Figure 1: Comparative view of clustering representations learned for a 2D maze (a) environment with spiral walls. The agent's location is marked by black-dot in the maze image. The clustering of representations learned via ACRO (b) and PCLaSt (c) are overlaid on the maze image.

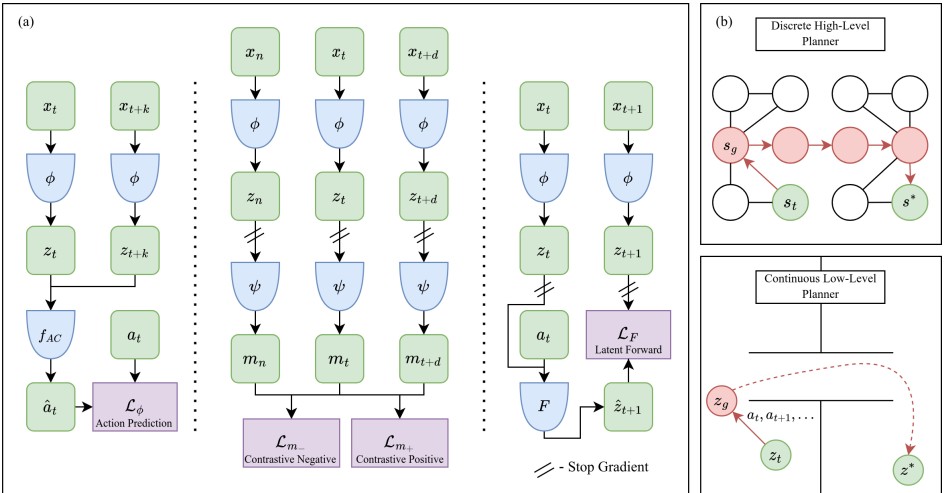

Figure 2: (a) Overview of the proposed method: The encoder, which maps observations $x$ to continuous latents $z$, is learned with a multi-step inverse model (left). A temporal contrastive objective is used to learn a metric space (middle), a forward model is learned in the latent space (right). (b) High-level and low-level planners. The high-level planner generates coarse goals ($s$) using a low-level continuous planner. The dashed line indicates the expected trajectory after $z_g$ is reached.

by obstacles are represented as close to each other without the boundary between them (see Fig. 1b). This inadequacy in capturing *all essential information useful for onward control tasks* is a drag on the efficacy of planning with deep RL algorithms despite their impressive showings in the last few years.

In this paper, we develop latent representations that accurately reflect states reachability in the quest towards sample-efficient planning from dense observations. We call this new approach plannable continuous latent states or PCLaST. Suppose that a latent representation, $\mathcal{Z}$, has been learned from a dense observation, $\mathcal{X}$, a PCLaST map from $\mathcal{Z}$ is learned via random data exploration. The map associates neighboring states together through this random exploration by optimizing a contrastive objective based on the likelihood function of a Gaussian random walk; The Gaussian is a reasonable model for random exploration *in the embedding space*. Figure 2 shows an overview of our approach, with a specific choice of the initial latent representation based on inverse dynamics.

We hypothesize that PCLaST representations are better aligned with the reachability structure of the environment. Our experiments validate that these representations improve the performance of reward-based and reward-free RL schemes. One key benefit of this representation is that it can be used to construct a discretized model of the environment and enable model-based planning to reach an arbitrary state from another arbitrary state. A discretized model (in combination with a simple local continuous planner) can also be used to solve more complex planning tasks that may require combinatorial solvers, like planning a tour across several states in the environment. Similarly to other latent state learning approaches, the learned representations can be used to drive more effective exploration of new states (Machado et al., 2017; Hazan et al., 2019; Jinnai et al., 2019; Amin et al., 2021). Since the distance in the PCLaST representation corresponds to the number of transitions between states, discretizing states at different levels of granularity gives rise to different levels of state abstraction. These abstractions can be efficiently used for hierarchical planning. In our experiments , we show that using multiple levels of hierarchy leads to substantial speed-ups in plan computation.

## 2 RELATED WORK

Our work relates to challenges in representation learning for forward/inverse latent-dynamics and using it for ad-hoc goal conditioned planning. In the following, we discuss each of these aspects.

**Representation Learning.** Learning representations can be decomposed into *reward-based* and *reward-free* approaches. The former involves both model-free and model-based methods. In model-free (Mnih et al., 2013), a policy is directly learned with rich observation as input. One can consider the penultimate layer as a latent-state representation. Model-based approaches like Hafner et al.

(2019a) learn policy, value, and/or reward functions along with the representation. These end-to-end approaches induce task-bias in the representation which makes them unsuitable for diverse tasks. In *reward-free* approaches, the representation is learned in isolation from the task. This includes model-based approaches (Ha & Schmidhuber, 2018), which learn a low-dimensional auto-encoded latent-representation. To robustify, contrastive methods (Laskin et al., 2020) learn representations that are similar across positive example pairs, while being different across negative example pairs. They still retain exogenous noise requiring greater sample and representational complexity. This noise can be removed (Efroni et al., 2021) from latent-state by methods like ACRO (Islam et al., 2022) which learns inverse dynamics (Mhammedi et al., 2023). These *reward-free* representations tend to generalize better for various tasks in the environment. The prime focus of discussed reward-based/free approaches is learning a representation robust to observational/distractor noise; whereas not much attention is paid to enforce the geometry of the state-space. Existing approaches hope that such geometry would emerge as a result of end-to-end training. We hypothesize lack of this geometry affects sample efficiency of learning methods. Temporal contrastive methods (such as HOMER Misra et al. (2020) and DRIML Mazoure et al. (2020)) attempt to address this by learning representations that discriminate among adjacent observations during rollouts, and pairs random observations Nair et al. (2022); Wang & Gupta (2015). However, this is still not invariant to exogenous information (Efroni et al., 2021).

**Planning.** Gradient descent methods abound for planning in learned latent states. For example, UPN (Srinivas et al., 2018) applies gradient descent for planning. For continuous latent states and actions, the cross-entropy method (CEM) (Rubinstein, 1999), has been widely used as a trajectory optimizer in model-based RL and robotics (Finn & Levine, 2017; Wang & Ba, 2019; Hafner et al., 2019b). Variants of CEM have been proposed to improve sample efficiency by adapting the sampling distribution of Pinneri et al. (2021) and integrating gradient descent methods (Bharadhwaj et al., 2020). Here, trajectory optimizers are recursively called in an online setting using an updated observation. This conforms with model predictive control (MPC) (Mattingley et al., 2011). In our work, we adopt a multi-level hierarchical planner that uses Dijkstra's graph-search algorithm (Dijkstra, 1959) for coarse planning in each hierarchy-level for sub-goal generation; this eventually guides the low-level planner to search action sequences with the learned latent model.

**Goal Conditioned Reinforcement Learning (GCRL).** In GCRL, the goal is specified along with the current state and the objective is to reach the goal in least number of steps. A number of efforts have been made to learn GCRL policies (Kaelbling, 1993; Nasiriany et al., 2019; Fang et al., 2018; Nair et al., 2018). Further, reward-free goal-conditioned (Andrychowicz et al., 2017) latent-state planning requires estimating the distance between the current and goal latent state, generally using Euclidean norm ($\ell_2$) for the same. However, it's not clear whether the learned representation is suitable for $\ell_2$ norm and may lead to infeasible/non-optimal plans; even if one has access to true state. So, either one learns a new distance metric (Tian et al., 2020; Mezghani et al., 2023) which is suitable for the learned representation or learns a representation suitable for the $\ell_2$ norm. In our work, we focus on the latter. Further, GCRL reactive policies often suffer over long-horizon problems which is why we use an alternate solution strategy focusing on hierarchical planning on learned latent state abstractions.

## 3 PCLaSt: Discovery, Representation, and Planning

In this section, we discuss learning the PCLaSt representation, constructing a transition model, and implementing a hierarchical planning scheme for variable-horizon state transitions.

### 3.1 Notations and Preliminaries.

We assume continuous state and action spaces throughout. Indices of time e.g. $t, t_0, \tau$ will always be integers and $\tau \gg t > t_0$. The Euclidean norm of a matrix, $X$, is denoted $\|X\|$. We adopt the exogenous block Markov decision process of (Efroni et al., 2021), characterized as the tuple $(\mathcal{X}, \mathcal{S}, \Xi, \mathcal{A}, T, q, R, \mu_0)$. Here, $\mathcal{X}, \mathcal{S}, \Xi$, and $\mathcal{A}$ are respectively the spaces of observations, true latent and exogenous noise states, and actions, respectively. The transition distribution is denoted $T(s_{t+1}, \xi_{t+1} \mid s_t, \xi_t, a_t)$ with true latent states $(s_t, s_{t+1}) \in S$, exogenous noise states $(\xi_t, \xi_{t+1}) \in \Xi$, and action $a_t \in \mathcal{A}$. At a fixed time $t$, the emission distribution over observations $x \in \mathcal{X}$ is $q(x \mid s, \xi)$, the reward function is $R : \mathcal{X} \times \mathcal{A} \to \mathbb{R}$, and $\mu_0(z, \xi)$ is the distribution over initial states, $z$. The agent interacts with its environment $\mathcal{E}$ generating latent state-action pairs $\{s_t, a_t\}_{t=0}^{\tau}$; here $s_t \sim \mu(x_t, \xi_t)$ for $x_t \sim q(\cdot \mid s_t)$. An encoder network maps observations $\{x_t\}_{t=0}^{\tau}$ to latent

states $\{s_t\}_{t=0}^{\mathcal{T}}$ while the transition function factorizes over actions and noise states as $T(s_{t+1}, \xi_{t+1} \mid s_t, \xi_t, a_t) = T_s(s'_{t+1}|s_t, a) \, T_\xi(\xi_{t+1}|\xi_t)$. The emission distribution enforces unique latent states from (unknown) mapped observations. We map each $\{x_t\}_{t=0}^{\mathcal{T}}$ to $\{s_t\}_{t=0}^{\mathcal{T}}$ under reachability constraints. We employ two encoder networks i.e. $\phi(x)$ and $\psi(s)$, and compose them as $\psi(\phi(x))$. In this manner, $\phi(x)$ eliminates exogenous noise whilst preserving latent state information, while $\psi(s)$, the PCLAST map, enforces the reachability constraints. The encoder $\phi(x)$ is based on the ACRO multi-step inverse kinematics objective of (Islam et al., 2022) whereas the encoder $\psi(s)$ uses a likelihood function in the form of a Gaussian random walk. Next, we discuss the learning scheme for the encoders and the PCLAST map, the forward model and planning schemes.

## 3.2 ENCODER DESCRIPTION.

The encoder is a mapping from observations to estimated (continuous) latent states, $\hat{z} \in \mathcal{Z}$, i.e., $\phi(x) : x \to \hat{z}$. Following Lamb et al. (2022); Islam et al. (2022), a multi-step inverse objective (reiterated in (1)) is employed to eliminate the exogenous noise. The loss (1) is optimized over the $f_{AC}$ network and the encoder $\phi(x)$ to predict $a_t$ from current and future state tuples,

$$\mathcal{L}_\phi(\phi, f_{AC}, x_t, a_t, x_{t+k}, k) = ||a_t - f_{AC}(\phi(x_t), \phi(x_{t+k}); k)||^2, \tag{1a}$$

$$\hat{\phi}(x) = \arg\min_{\phi \in \Phi} \min_{f_{AC}} \mathbb{E}_t \mathbb{E}_k \, \mathcal{L}_\phi\left(\phi, f_{AC}, x_t, a_t, x_{t+k}, k\right), \tag{1b}$$

where $f_{AC} \in \mathcal{F}_{AC}$ is the action predictor, $t \sim U(1, \mathcal{T})$ is the index of time, and $k \sim U(1, K_{max})$ is the amount of look-ahead steps. We uniformly sample $k$ from the interval $[1, K_{max}]$, where $K_{max}$ is the diameter of the control-endogenous MDP (Lamb et al., 2022). The encoder $\phi(x)$, as a member of a parameterized encoders family $\Phi$, maps images, $x$, to a low-dimensional latent representation, $s$. A fully-connected network $f_{AC} : \mathcal{Z} \times \mathcal{Z} \times [K_{max}] \to \mathcal{A}$, belonging to a parameterized family $f_{AC} \in \mathcal{F}_{AC}$, is optimized alongside $\phi(x)$. This predicts the action, $a$, from a concatenation of representation vectors $z_t$, $z_{t+k}$, and embedding of $k$ from (1). Intuitively, the action-predictor $f$ models the conditional probability over actions $p(a_t|\phi(x_t), \phi(x_{t+k}); k)$[1].

## 3.3 LEARNING THE PCLAST MAP.

While the encoder $\phi(x)$ is designed to filter out the exogenous noise, it does not lead to representations that reflect the reachability structure (see Fig. 1b). To enforce states' reachability, we learn a map $\psi(x)$, which associates nearby states based on transition deviations. Learning $\psi(x)$ is inspired from local random exploration that enforces a Gaussian random walk in the embedding space. This allows states visited in fewer transitions to be closer to each other.

We employed a Gaussian random walk with variance $\sigma I$ (where $I$ is an identity matrix) for $k$ steps to induce a conditional distribution, given as $(s_{t+k}|s_t) \propto exp\left\{-\frac{||s_{t+k}-s_t||^2}{2k\sigma^2}\right\}$. Instead of optimizing $\psi(x)$ to fit this likelihood directly, we fit a contrastive version, based on the following generative process for generating triples $y, s_t, s_{t+1}$. First, we flip a random coin whose outcome $y \in \{0, 1\}$; and then predict $y$ using $s_t$ and $s_{t+k}$. This objective takes the form,

$$\mathbb{P}_k(y = 1|s_t, s_{t+k}) = \sigma(\beta - \alpha||s_t - s_{t+k}||), \tag{2}$$

and it is sufficient as shown in Appendix B. Another encoder $\psi(x) : \mathcal{Z} \to \mathcal{Z}'$ estimates the states reachability so that the output of $\psi(x)$ prescribes that close-by points be locally reachable with respect to the latent agent-centric latent state. $\mathcal{Z}'$ is learned latent state with true state reachability property.

A contrastive learning loss $\mathcal{L}_\psi$ is minimized to find $\psi(x)$ along with the scaling parameters $\alpha$ and $\beta$ by averaging over the expected loss as

$$\mathcal{L}_{m_+}(\psi, \hat{z}_A, \hat{z}_B, \alpha, \beta) = -\log(\sigma(e^\alpha - e^\beta ||\psi(\hat{z}_A) - \psi(\hat{z}_B)||^2)), \tag{3a}$$

$$\mathcal{L}_{m_-}(\psi, \hat{z}_A, \hat{z}_B, \alpha, \beta) = -\log(1 - \sigma(e^\alpha - e^\beta ||\psi(\hat{z}_A) - \psi(\hat{z}_B)||^2)), \tag{3b}$$

$$\mathcal{L}_\psi(\psi, \phi, \alpha, \beta, x_t, x_{t+d}, x_r) = \mathcal{L}_{m_+}(\psi, \phi(x_t), \phi(x_{t+d}), \alpha, \beta) + \mathcal{L}_{m_-}(\psi, \phi(x_t), \phi(x_r), \alpha, \beta), \tag{3c}$$

$$\psi, \alpha, \beta = \arg\min_{\substack{\psi \in \Psi, \\ \alpha, \beta \in \mathbb{R}}} \mathbb{E}_{t,r} \mathbb{E}_d \, \mathcal{L}_\psi(\psi, \phi, \alpha, \beta, x_t, x_{t+d}, x_r), \tag{3d}$$

---

[1]We assume that this conditional distribution is Gaussian with a fixed variance.

where $t \sim U(1, \mathcal{T})$, $r \sim U(1, \mathcal{T})$, $d \sim U(1, d_m)$ for a hyperparameter $d_m$, and $e^\alpha$ and $e^\beta$ provide smoothly enforced value greater than 0. Positive examples are drawn for the contrastive objective uniformly over $d_m$ steps, and negative examples are sampled uniformly from a data buffer

### 3.4 Learning a latent forward model and compositional planning.

We now describe the endowment of learned latent representations with a forward model, which is then used to construct a compositional multi-layered planning algorithm.

**Forward Model.** A simple latent forward model $F : \mathcal{Z} \times \mathcal{A} \to \mathcal{Z}$ estimates the latent forward dynamics $\phi(x_{t+1}) \approx F(\phi(x_t), a_t)$. The forward model $F$ is parameterized as a fully-connected network of a parameterized family $\mathcal{F}$, optimized with a prediction objective,

$$\mathcal{L}_F(F, x_t, a_t, x_{t+1}) = ||\phi(x_{t+1}) - F(\phi(x_t), a_t)||^2, \tag{4a}$$

$$F = \arg \min_{F \in \mathcal{F}} \mathbb{E}_t \mathcal{L}_F(F, \phi(x_t), a_t, \phi(x_{t+1})). \tag{4b}$$

**High-Level Planner.** Let $z_t = \phi(x_t)$ denote the latent state. In the planning problem, we aim to navigate the agent from an initial latent state $z_{init}$ to a target latent state $z_{goal}$ following the latent forward dynamics $z_{t+1} = F(z_t, a_t)$. Since $F$ is highly nonlinear, it presents challenges for use in global planning tasks. Therefore, we posit that a hierarchical planning scheme with multiple abstraction layers can improve the performance and efficacy of planning by providing waypoints for the agent to track using global information of the environment.

To find a waypoint $z^*$ in the latent space, we first divide the latent space into $C$ clusters by applying k-means to an offline collected latent states dataset, and use the discrete states $\{s_i\}_{i=1}^C$ to denote each cluster. An abstraction of the environment is given by a graph $\mathcal{G}$ with nodes $\{s_i\}_{i=1}^C$ and edges defined by the reachability of each cluster, i.e., an edge from node $s_i$ to node $s_j$ is added to the graph if there are transitions of latent states from cluster $s_i$ to cluster $s_j$ in the offline dataset. On the graph $\mathcal{G}$, we apply Dijkstra's shortest path algorithm (Dijkstra, 1959) to find the next cluster the agent should go to and choose the center latent state of that cluster as the waypoint $z^*$. This waypoint is passed to a low-level planner to compute the action.

**Low-Level Planner.** Given the current latent state $z_0$ and the waypoint $z^*$ to track, the low-level planner finds the action to take by solving a trajectory optimization problem using the cross-entropy method (CEM) (De Boer et al., 2005). The details are shown in Appendix D.

**Multi-Layered Planner.** To improve the efficiency of finding the waypoint $z^*$, we propose to build a hierarchical abstraction of the environment such that the high-level planner can be applied at different levels of granularity, leading to an overall search time reduction of Dijkstra's shortest path algorithm. Let $n \geq 2$ denote the number of abstraction levels [2] and a higher number means coarser abstraction. At level $2 \leq i \leq n$, we partition the latent space into $C_i$ clusters using k-means, and we have $C_2 > C_3 > \cdots > C_n$. For each abstraction level, we construct the discrete transition graph $\mathcal{G}_i$ accordingly, which is used to search for the waypoint $z^*$ with increasing granularity as shown in Algorithm 1. This procedure guarantees that the start and end nodes are always a small number of hops away in each call of Dijkstra's algorithm. In Section 4.4, our experiments show that multi-layered planning leads to a significant speedup compared with using only the finest granularity.

## 4 Experiments

In this section, we address the following questions via experimentation over environments of different complexities: 1) Does the PCLAST representation lead to performance gains in reward-based and reward-free goal-conditioned RL settings? 2) Does increasing abstraction levels lead to more computationally efficient and better plans? 3) What is the effect of PCLAST map on abstraction?

### 4.1 Environments

We consider three categories of environments for our experiments and discuss them as follows:

**Maze2D - Point Mass.** We created a variety of 2D maze point-mass environments with continuous actions and states. The environments are comprised of different wall configurations with the goal of

---

[2] When $n = 1$, we only apply the low-level planner without searching for any waypoint.

---

**Algorithm 1:** Multi-Layered planner

---

**Data:** Current observation $x_t$, goal observation $x_{goal}$, planning horizon $T$, encoder $\phi(\cdot)$, latent
forward dynamics $F(\cdot, \cdot)$, multi-layer discrete transition graphs $\{\mathcal{G}_i\}_{i=2}^n$.

**Result:** Action sequence $\{a_i\}_{i=0}^{T-1}$.

1 Compute current continuous latent state $z_t = \phi(x_t)$ and target latent state $z^* = \phi(x_{goal})$.

2 //* See Appendix D for details of high-level planner and low-level planner.

3 **for** $i = n, n-1, \cdots, 2$ **do**

4     $z^*$ = high-level planner($z_t, z^*, \mathcal{G}_i$).       ▷ Update waypoint using a hierarchy of abstraction.

5 **end**

6 $\{a_i\}_{i=0}^{T-1}$ = low-level planner($z_t, z^*, T, F$);       ▷ Solve the trajectory optimization problem.

---

navigating a point-mass. The size of the grid is $(100 \times 100)$ and each observation is a 1-channel image
of the grid with "0" marking an empty location and "1" marking the ball's coordinate location $(x, y)$.
Actions comprise of $(\Delta x, \Delta y)$ and specify the coordinate space change by which the ball should
be moved. This action change is bounded by $[-0.2, 0.2]$. There are three different maze variations:
MAZE-HALLWAY, MAZE-SPIRAL, and MAZE-ROOMS whose layouts are shown in Fig. 3(a,b and c).
Further, we have dense and sparse reward variants for each environment, details of which are given
in Appendix C.1. We created an offline dataset of 500K transitions using a random policy for each
environment which gives significant coverage of the environment's state-action space.

**Robotic-Arm.** We extended our experiments to the *Sawyer-Reach* environment of Nair et al. (2018)
(shown in Fig. 3d). It consists of a 7 DOF robotic arm on a table with an end-effector. The end-effector
is constrained to move only along the planar surface of the table. The observation is a $(84 \times 84)$ RGB
image of the top-down view of the robotic arm and actions are 2 dimensional continuous vectors
that control the end-effector coordinate position. The agent is tested on its ability to control the
end-effector to reach random goal positions. The goals are given as images of the robot arm in the
goal state. Similar to maze2d environment, we generate an offline dataset of 20K transitions using
rollouts from a random policy. Likewise to maze, it has dense and sparse reward variant.

**Exogenous Noise Mujoco.** We adopted control-tasks *"Cheetah-Run"* and *"Walker-walk"* from visual-
d4rl (Lu et al., 2022) benchmark which provides offline transition datasets of various qualities. We
consider *"medium, medium-expert, and expert"* datasets. The datasets include high-dimensional agent
tracking camera images. We add exogenous noise to these images to make tasks more challenging,
details are given in the Appendix C.2. The general objective in these task is to keep agent alive and
move forward, while agent feed on exogenous noised image.

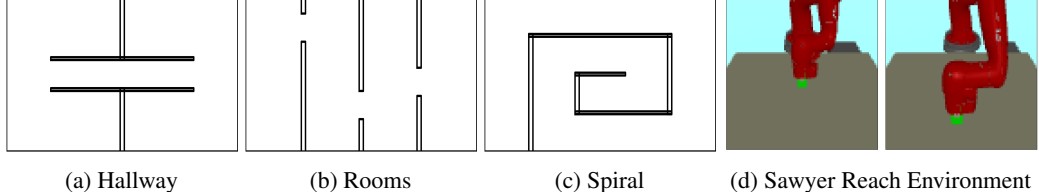

| (a) Hallway | (b) Rooms | (c) Spiral | (d) Sawyer Reach Environment |

Figure 3: Environments: (a), (b) and (c) show different wall-configurations of *Maze2d* environment
for point-mass navigation task and (d) shows top-down view of robot-arm environment with the task
of reaching various goal positions in 2D-planar space.

## 4.2 IMPACT OF REPRESENTATION LEARNING ON GOAL CONDITIONED RL

We investigate the impact of different representations on performance in goal-conditioned model-free
methods. First, we consider methods which use explicit-reward signal for representation learning. As
part of this, we trained goal-conditioned variant of PPO (Schulman et al., 2017) on each environment
with different current state and goal representation methods. This includes : (1) Image representation
for end-to-end learning, (2) ACRO representation Islam et al. (2022), and (3) PCLAST representation.
For (1), we trained PPO for 1 million environment steps. For (2) and (3), we first trained representation
using offline dataset and then used frozen representation with PPO during online training for 100K
environment steps only. In the case of *Sawyer-Reach*, we emphasize the effect of limited data and

reserved experiments to 20K online environment steps. We also did similar experiment with offline CQL (Kumar et al., 2020) method with pre-collected dataset.

Secondly, we consider RL with Imagined Goals (RIG) (Nair et al., 2018) a method which *doesn't need an explicit reward signal* for representation learning and planning. It is an online algorithm which first collects data with simple exploration policy. Thereafter, it trains an embedding using VAE on observations (images) and fine-tunes it over the course of training. Goal conditioned policy and value functions are trained over the VAE embedding of goal and current state. The reward function is the negative of $\ell_2$ distance in the latent representation of current and goal observation. In our experiments, we consider pre-trained ACRO and PCLaST representation in addition to default VAE representation. Pre-training was done over the datasets collected in Section 4.1.

Our results in Table 1 show PPO and CQL have poor performance when using direct images as representations in maze environments. However, ACRO and PCLaST representations improve performance. Specifically, in PPO, PCLaST leads to significantly greater improvement compared to ACRO for maze environments. This suggests that enforcing a neighborhood constraint facilitates smoother traversal within the latent space, ultimately enhancing goal-conditioned planning. PCLaST in CQL gives significant performance gain for *Maze-Hallway* over ACRO; but they remain within standard error of each other in *Maze-Rooms* and *Maze-Spiral*. Generally, each method does well on *Sawyer-Reach* environment. We assume it is due to lack of obstacles which allows a linear path between any two positions easing representation learning and planning from images itself. In particular, different representations tend to perform slightly better in different methods such as ACRO does better in PPO (sparse), PCLaST does in CQL, and image itself does well in PPO(dense) and RIG.

### 4.3 IMPACT OF PCLaST ON STATE-ABSTRACTION

We now investigate the quality of learned latent representations by visualizing relationships created by them across true-states. This is done qualitatively by clustering the learned representations of observations using *k*-means. Distance-based planners use this relationship when traversing in latent-space. In Fig. 4 ($2^{nd}$ row), we show clustering of PCLaST representation of offline-observation datasets for maze environments. We observe clusters having clear separation from the walls. This implies only states which are reachable from each other are clustered together. On the other hand, with ACRO representation in Fig. 4 (3rd row), we observe disjoint sets of states are categorized as single cluster such as in cluster-10 (orange) and cluster-15 (white) of Maze-Hallway environment. Further, in some cases, we have clusters which span across walls such as cluster-14 (light-pink) and cluster-12 (dark-pink) in Maze-spiral environment. These disjoint sets of states violate a planner's state-reachability assumption, leading to infeasible plans.

| Method | Reward type | Hallway | Rooms | Spiral | Sawyer-Reach |
|---|---|---|---|---|---|
| PPO | Dense | $6.7 \pm 0.6$ | $7.5 \pm 7.1$ | $11.2 \pm 7.7$ | $\mathbf{86.00 \pm 5.367}$ |
| PPO + ACRO | Dense | $10.0 \pm 4.1$ | $23.3 \pm 9.4$ | $23.3 \pm 11.8$ | $84.00 \pm 6.066$ |
| PPO + PCLaST | Dense | $\mathbf{66.7 \pm 18.9}$ | $\mathbf{43.3 \pm 19.3}$ | $\mathbf{61.7 \pm 6.2}$ | $78.00 \pm 3.347$ |
| PPO | Sparse | $1.7 \pm 2.4$ | $0.0 \pm 0.0$ | $0.0 \pm 0.0$ | $68.00 \pm 8.198$ |
| PPO + ACRO | Sparse | $21.7 \pm 8.5$ | $5.0 \pm 4.1$ | $11.7 \pm 8.5$ | $\mathbf{92.00 \pm 4.382}$ |
| PPO + PCLaST | Sparse | $\mathbf{50.0 \pm 18.7}$ | $\mathbf{6.7 \pm 6.2}$ | $\mathbf{46.7 \pm 26.2}$ | $82.00 \pm 5.933$ |
| CQL | Sparse | $3.3 \pm 4.7$ | $0.0 \pm 0.0$ | $0.0 \pm 0.0$ | $32.00 \pm 5.93$ |
| CQL + ACRO | Sparse | $15.0 \pm 7.1$ | $\mathbf{33.3 \pm 12.5}$ | $\mathbf{21.7 \pm 10.3}$ | $68.00 \pm 5.22$ |
| CQL + PCLaST | Sparse | $\mathbf{40.0 \pm 0.5}$ | $23.3 \pm 12.5$ | $20.0 \pm 8.2$ | $\mathbf{74.00 \pm 4.56}$ |
| RIG | None | $0.0 \pm 0.0$ | $0.0 \pm 0.0$ | $3.0 \pm 0.2$ | $\mathbf{100.0 \pm 0.0}$ |
| RIG + ACRO | None | $\mathbf{15.0 \pm 3.5}$ | $4.0 \pm 1.$ | $\mathbf{12.0 \pm 0.2}$ | $100.0 \pm 0.0$ |
| RIG + PCLaST | None | $10.0 \pm 0.5$ | $4.0 \pm 1.8$ | $10.0 \pm 0.1$ | $90.0 \pm 5$ |
| H-Planner + PCLaST | None | $\mathbf{97.78 \pm 4.91}$ | $\mathbf{89.52 \pm 10.21}$ | $\mathbf{89.11 \pm 10.38}$ | $95.0 \pm 1.54$ |

Table 1: Impact of different representations on policy learning and planning. The numbers represent mean and standard error of the percentage success rate of reaching goal states, estimated over 5 random seeds. RIG and H-planner do not use an external reward signal. In H-planner, we use $n = 5$ abstraction levels. Highlighted in bold font are the methods with the best mean performance in each task.

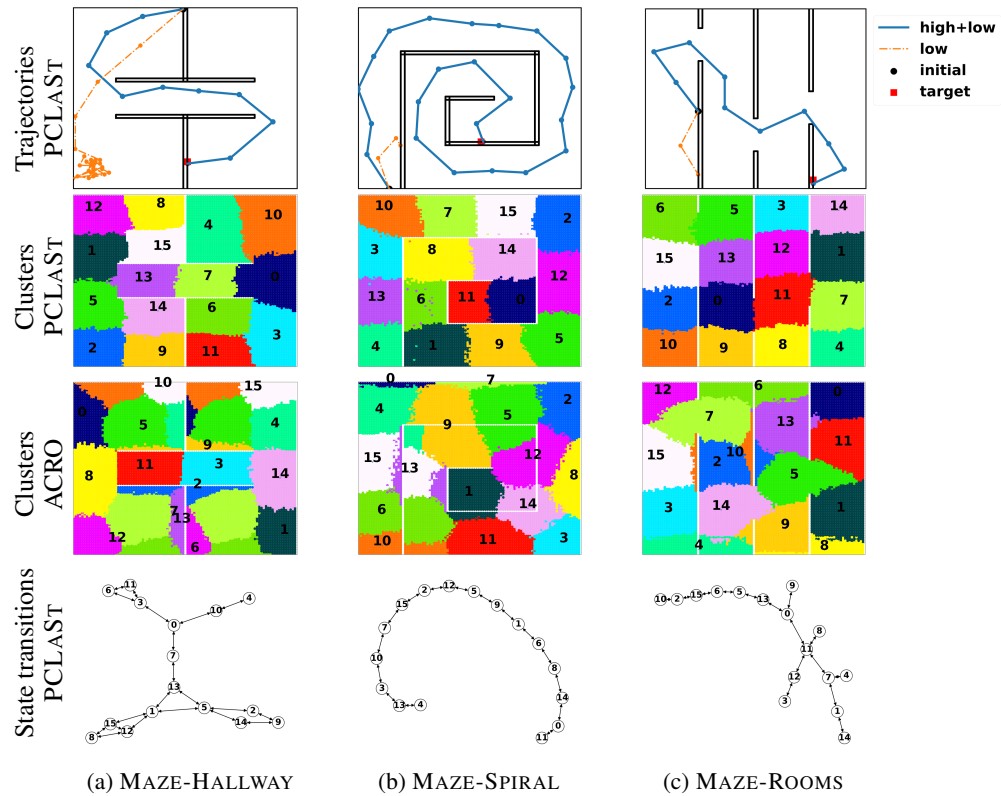

(a) MAZE-HALLWAY      (b) MAZE-SPIRAL      (c) MAZE-ROOMS

Figure 4: Clustering, Abstract-MDP, and Planning are shown for Maze environments in each column. In the first row, we show the maze configuration and the executed trajectories of the agent from the initial location (black) to the target location (red) using *high+low* planners (blue) and just low-level planners (orange). In the second and third rows, we show k-means "($k = 16$)" clustering of latent states learned by PCLAST and ACRO, respectively. Finally, in the fourth row, we show the abstract transition model of the discrete states learned by PCLAST (2nd row) which captures the environment's topology.

## 4.4 MULTI-LEVEL ABSTRACTION AND HIERARCHICAL PLANNING

In Section 4.2, we found PCLAST embedding improves goal-conditioned policy learning. However, reactive policies tend to generally have limitations for long-horizon planning. This encourages us to investigate the suitability of PCLAST for *n-level* state-abstraction and hierarchical planning with Algorithm 1 which holds promise for long-horizon planning. Abstractions for each level are generated using k-means with varying $k$ over the PCLAST embedding as done in Section 4.3.

For simplicity, we begin by considering *2*-level abstractions and refer to them as high and low levels. In Fig. 4, we show that the learned clusters of high-level in the second row and the abstract transition models of the discrete states in the last row, which match with the true topology of the mazes. Using this discrete state representation, MPC is applied with the planner implemented in Algorithm 1. Our results show that in all cases, our hierarchical planner (*high + low*) generates feasible and shortest plans (blue-line), shown in the top row of the Fig. 4. As a baseline, we directly evaluate our *low-level* planner (see the orange line) over the learned latent states which turns out to be failing in all the cases due to the long-horizon planning demand of the task and complex navigability of the environment.

**Increasing Abstraction Levels.** We investigate planning with multiple abstraction levels and consider $n \in \{2, 3, 4, 5\}$. Performance score for "$n = 5$" are reported in Table 1 (lowest-row). These abstractions help us create a hierarchy of graphs that describes the environment. In Fig. 5, we use $k = \{32, 16, 8, 4\}$ for $n = \{2, 3, 4, 5\}$ abstraction levels, respectively, and show graph-path for each abstraction-level for planning between two locations in the maze-environment. This multi-level planning gives a significant boost to planning performance as compared to our model-free baselines.

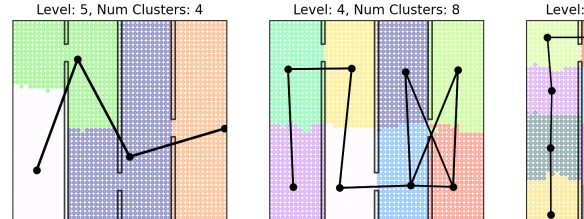
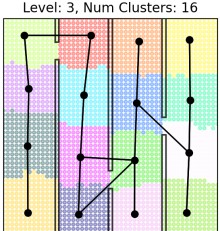
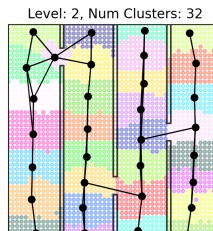

Figure 5: Visualization of hierarchical graphs in the *Maze2d-hallway* environment. At every level, num clusters is the $k$ used for clustering. The transition graph (in black) constructed from the cluster centers is superimposed over the environment. The hierarchy is indexed from 1 for the lowest level and increases the higher the level.

Similarly, we observe $3.8\times$ *computational time efficiency improvement in planning with "$n = 5$" (0.07 ms) as compared to "$n = 2$" (0.265 ms) abstraction levels*. However, no significant performance gains were observed. We assume this is due to the good quality of temporal abstraction at just $n = 2$ which leads to the shortest plans and increasing the levels just helps to save on computation time. However, for more complex tasks, increasing the abstraction levels may further increase the quality of plans.

### 4.5 EXOGENOUS-NOISE OFFLINE RL EXPERIMENTS

Finally, we evaluate PCLAST exclusively on exogenous noised control environments described in Section 4.1. We follow the same experiment setup as done by Islam et al. (2022) and consider ACRO Islam et al. (2022), DRIML Mazoure et al. (2020), HOMER Misra et al. (2020), CURL Laskin et al. (2020) and 1-step inverse model Pathak et al. (2017) as our baselines. We share results for *"Cheetah-Run"* with *"expert, medium-expert, and medium"* dataset in Fig. 6. It shows PCLAST helps gain significant performance over the baselines (Islam et al., 2022). Extended results for *"Walker-Walk"* with similar performance trends are shown in Fig. 9(Appendix).

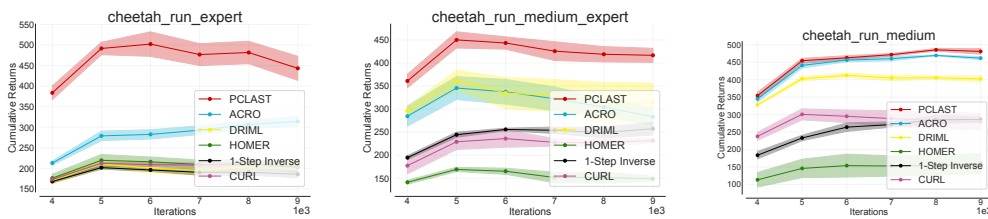

Figure 6: Comparisons of PCLAST in *Cheetah-Run* exogenous-noise environment with several other baselines.

## 5 SUMMARY

Learning competent agents to plan in environments with complex sensory inputs, exogenous noise, non-linear dynamics, along with limited sample complexity requires learning compact latent-representations which maintain state affordances. Our work introduces an approach which learns a representation via a multi-step inverse model and temporal contrastive loss objective. This makes the representation robust to exogenous noise as well as retains local neighborhood structure. Our diverse experiments suggest the learned representation is better suited for reactive policy learning, latent-space planning as well as multi-level abstraction for computationally-efficient hierarchical planning.

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

## A    INTUITIVE ARGUMENT: LEARNING LNS VIA CONTRASTIVE LEARNING

In the previous section we introduced a contrastive learning objective which allows us to learn the underlying LNS in a scalable and differentiable way (see equations (3a)-(3)). Next, we elaborate on the intuition that resulted in this objective. We show that the newly introduced temporal contrastive loss can be derived assuming a natural diffusion process on the underlying dynamics.

Consider a simple discrete time multi-dimensional Brownian motion. The conditional probability to observe the process at position $z'$ at time step $k$ conditioning on it starting from $z$ at time step $k = 0$, denoted as $\mathbb{P}_k(z'|z_0 = z)$, is given as (see Durrett (2019), Section 7):

$$\mathbb{P}_k(z'|z_0 = z) \propto \exp\left(\frac{||z - z'||^2}{\sigma_0 k}\right).$$

With this fact in mind we can study the distribution of the contrastive learning process which outputs the tuple $(y, z, z')$. A simple way to define this process is as follows: sample $z$ uniformly from $\mathcal{Z}$, sample $z'$ after $k$ steps where $\bar{z} \sim \mathbb{P}_k(\cdot|z_0 = z)$, independently, sample $y \sim \text{Bernoulli}(0.5)$, if $y = 1$ set $z' = \bar{z}$ and otherwise sample $z'$ uniformly at random. We refer to this process as the *Contrastive Learning(CL) generating process*. The following can be readily derived from these assumptions (see proof in Appendix B).

**Proposition 1.** *Assume the tuple $(y, z, z')$ is sampled via the CL generating process described above. Then, $\mathbb{P}_k(y = 1 \mid z, z') = \text{sigmoid}(c - b||z - z'||^2)$, where $\text{sigmoid}(x) = \exp(x)/(\exp(x) + 1)$.*

Although this generating process does not take into account the geometry of the underlying space and subtle intricacies of the environment, for small time scales this process can capture the dynamics to a reasonable degree. Further, the contrastive learning objective follows directly from Proposition 1: this objective can be interpreted as a log-likelihood learning procedure of the CL generating process.

## B   BAYES SOLUTION OF CONTRASTIVE LOSS

**Proposition 1.** *Assume the tuple $(y, z, z')$ is sampled via the CL generating process described above. Then, $\mathbb{P}_k(y = 1 \mid z, z') = \text{sigmoid}(c - b||z - z'||^2)$, where $\text{sigmoid}(x) = \exp(x)/(\exp(x) + 1)$.*

*Proof.* The proof following by direct analysis of the conditional probability distribution together with the assumption of the CL generating process, i.e., the underlying Brownian motion.

$$\mathbb{P}_k(y = 1|z, z') = \frac{\mathbb{P}_k(z|z', y = 1)\mathbb{P}_k(y = 1|z)}{\mathbb{P}_k(z'|z', y = 1)\mathbb{P}_k(y = 1|z) + \mathbb{P}_k(z'|z, y = 0)\mathbb{P}_k(y = 0|z)}$$

$$= \frac{\mathbb{P}_k(z|z', y = 1)}{\mathbb{P}_k(z'|z', y = 1) + \mathbb{P}_k(z'|z, y = 0)}$$

$$= \frac{\mathbb{P}_k(z|z', y = 1)}{\mathbb{P}_k(z'|z', y = 1) + 1/|\mathcal{Z}|}$$

$$= \frac{C \exp\left(-\frac{||z-z'||^2}{\sigma_0 k}\right)}{C \exp\left(\frac{-||z-z'||^2}{\sigma_0 k}\right) + 1/|\mathcal{Z}|}$$

$$= \frac{\exp\left(\log(C|\mathcal{Z}|) - \frac{||z-z'||^2}{\sigma_0 k}\right)}{\exp\left(\log(C|\mathcal{Z}|) - \frac{||z-z'||^2}{\sigma_0 k}\right) + 1}$$

$$= \text{sigmoid}(c - b||z - z'||^2).$$

The first relation is an application of Bayes' rule; the second relation follows since $y \sim \text{Bernoulli}(0.5)$ sampled independently from $z$ and $z'$; the third relation holds since $z'$ is assumed to be sampled uniformly when $y = 0$; the forth relation holds by the Brownian motion assumption (where $C$ is a positive constant); the sixth relation holds by defining $b = 1/(\sigma_0 k)$ and $c = \log(C|\mathcal{Z}|)$.

□

## C   EXPERIMENTS

Over here, we discuss our environment in details.

### C.1   MAZE-2D POINT MASS

**Environment setup** The state $s_t$ of the point-mass experiment is the 2D position of a point and the action $a_t$ is the position displacement, i.e., $s_{t+1} = s_t + a_t$. The action is bounded by $||a_t||_\infty \leq 0.2$. In the presence of obstacles, the point mass starting from $s_t$ is moved along the direction of $a_t$ until it collides with an obstacle. Further, we have two reward variants for each maze : 1) *Dense-reward* and 2) *Sparse-reward*. In the dense case, the agent receives a reward for the first time it crosses a particular distance threshold from the goal. Specifically, if $d_g$ is the distance to the goal, the agent receives a reward $r$ encouraging it to go closer to the goal. The thresholds for reaching the goal, as well as the corresponding reward values are given below.

$$\begin{aligned} r &= 0.25 & &\text{if this is the first time } d_g < 0.1 \\ &= 0.5 & &\text{if this is the first time } d_g < 0.05 \\ &= 1 & &\text{if } d_g < 0.03 \end{aligned}$$

In the sparse setting, the agent receives a reward of 1 when it's within a distance of 0.03 to the goal state. For evaluation, we randomize the start and goal states from across the maze so as to test the agent's ability to reach diverse goals.

As shown in Fig. 4 in the main paper, we consider three environments with distinct layouts of obstacles. In each of these environments, a dataset of 500K samples is collected using random actions.

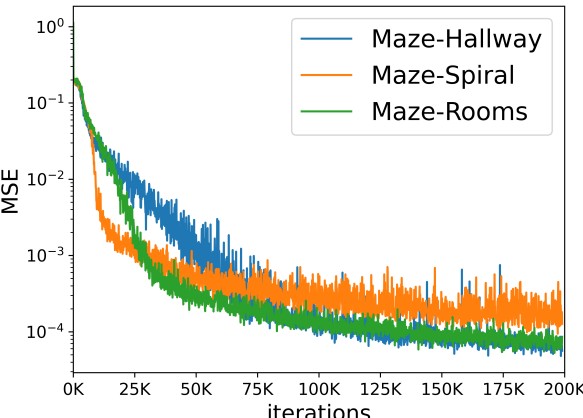

Figure 7: Mean square error of predicting the true state using the learned latent states during the training.

An instance of the dataset has *<obs-image, state, action, next-state, next-obs-image>*, where *state* and *next-state* are the coordinates of the agent in the maze and represent the true environment state as the obstacles are fixed. We use the transaction data to train the latent dynamics and extract an abstract transaction model using k-means clustering over the latent states with the continuous actions discretized with a resolution of $0.01$ for identifying transitions of the latent states across different clusters.

**Quality of the latent representation** Here we demonstrate the quality of the learned latent representation. Since, in the 2D point-mass experiment, we have access to the true state of the environment, we train a feed-forward network to predict the true state from the learned latent state. In Fig. 7, we show the regression error of predicting the true state from the latent state. A significant low error implies that the learned latent state captures information about the true state.

**Transition model generation** In constructing the transition models between the clusters of latent states, we filter out the infrequent transitions to avoid giving hard-to-reach goals to the low-level planner. For the cluster $s_i$, let $s_{i_1}, s_{i_2}, \cdots, s_{i_N}$ denote the $N$ clusters such that there is at least a state transaction from $s_i$ to $s_{i_j}$ in the collected samples. This means it is feasible to move the point mass from $s_i$ to $s_{i_j}$. Let $0 < p_{i_j} \leq 1$ for $j = 1, \cdots, N$ denote the ratio of the number of transactions from $s_i$ to $s_{i_j}$ to the total number of outward transactions from $s_i$. We observe that if $p_{i_j}$ is small, the following issues may happen: (i) The transition from $s_i$ to $s_{i_j}$ is caused by the clustering errors and does not give a feasible transaction of the agent in practice. (ii) Even if such a transition is feasible, it is difficult for the low-level planner to find such a path as indicated by the sparsity of such transitions in the collected dataset. Therefore, in generating the transition models, we add the edge $s_i \rightarrow s_{i_j}$ to the graph only when $p_{i_j}$ is large enough. Without loss of generality, assume that $p_{i_j}$ for $j = 1, \cdots, N$ have been arranged in descending order. Motivated by the nucleus sampling, we choose $N^* = \arg\min_k \sum_{j=1}^{k} p_{i_j} \geq 0.9$ and only add the edges $s_i \rightarrow s_{i_j}$ for $j = 1, \cdots, N^*$ to the graph. In this way, the sparse transactions are filtered out.

### C.2 EXOGENOUS NOISE OFFLINE RL EXPERIMENTS

We add exogenous noise by sampling 3 observations from "random" quality dataset and adding them around the main observation as shown in Figure 8. This creates a $4 \times 4$ exogenous observation from the offline dataset. This is same setup as from Islam et al. (2022). In our experiments, we have the controllable environment in one corner of the grid, and 3 other uncontrollable environments, taken from a random dataset, placed randomly in the $4 \times 4$ grid. Figure 9 shows additional results in the offline RL experimental setup.

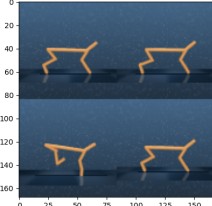 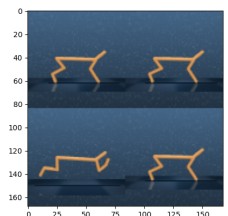 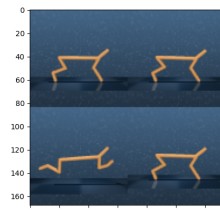

Figure 8: Illustration of the exogenous observations from the offline dataset, where the exogenous uncontrollable noise is placed in a $4 \times 4$ grid beside the controllable environment. We follow a setup similar to Islam et al. (2022), where both the endogenous and the exogenous observations are taken from the same domain (e.g Cheetah-Run domain as demonstrated here), except that the controllable observation may come from an expert or medium-expert dataset, whereas all the exogenous observations are taken from a random dataset.

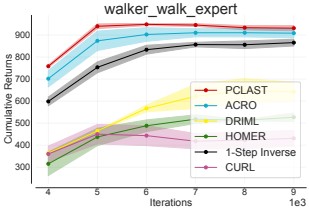 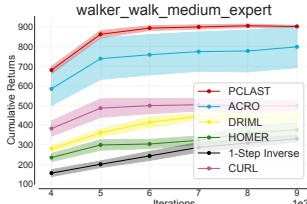 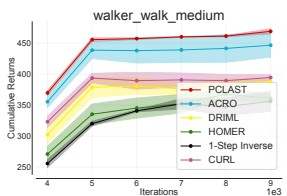

Figure 9: Comparisons of PCLAST with several other baselines, following the $4 \times 4$ exo-grid observation space offline RL setup from Islam et al. (2022). Experimental results on the Walker-Walk domain.

## D  MULTI-LAYERED PLANNER IMPLEMENTATION DETAILS

### D.1  HIGH-LEVEL PLANNER

Given the current latent state $z_t$ and the target latent state $z^*$, the high-level planner aims to find an intermediate waypoint $\tilde{z}$ such that the low-level planner can effectively track $\tilde{z}$. The search for the waypoint is based on the discrete abstraction of the environment which is given by the graph $\mathcal{G}$ as described in Section 3.4. We denote $\phi_d(\cdot)$ as the node (or cluster) membership function for the graph abstraction $\mathcal{G}$ such that $\phi_d(z)$ returns the node in $\mathcal{G}$ for any latent state $z \in \mathcal{Z}$. The high-level planner is outlined in Algorithm 2

---

**Algorithm 2:** High-level planner

**Data:** Current latent state $z_t$, target latent state $z^*$, graph abstraction $\mathcal{G}$. $\triangleright$ The node membership function $\phi_d(\cdot)$ is given by $\mathcal{G}$.

**Result:** Waypoint $\tilde{z}$.

1 Find the discrete latent states $s_t = \phi_d(z_t)$ and $s^* = \phi_d(z^*)$.
2 **if** $s_t \neq s^*$ **then**
3     $\tilde{s} = \text{Dijkstra}(s_t, s^*, \mathcal{G})$           $\triangleright \tilde{s}$ is the next node on the shortest path from $s_t$ to $s^*$.
4     set $\tilde{z}$ as the center of the cluster of $\tilde{s}$.
5 **else**
6     $\tilde{s} = s^*, \tilde{z} = z^*$.
7 **end**
8 Return $\tilde{z}$.

---

---

**Algorithm 3:** Cross-entropy method

---

**Data:** Number of iteration $N$, number of samples $M$ each iteration, and parameter $K$.

**Result:** Action sequence $\{a_i^*\}_{i=0}^{T-1}$.

1 Initialize a multivariate Gaussian distribution $\mathcal{N}(\mu^{(0)}, \Sigma^{(0)})$ with mean $\mu^{(0)} = \mathbf{0}$ and covariance matrix $\Sigma^{(0)} = I$.

2 **for** $j = 0, \cdots, N-1$ **do**

3      Sample $M$ action sequences $\{a_i^{(m)}\}_{i=1}^{T-1}$ from $\mathcal{N}(\mu^{(j)}, \Sigma^{(j)})$ for $m = 1, \cdots, M$.

4      For each action sequence $\{a_i^{(m)}\}_{i=1}^{T-1}$, evaluate the rendered cost $J^{(m)}$ of Problem (5).

5      Select the $K$ smallest costs from $\{J^{(m)}\}_{m=1}^{M}$ and the corresponding action sequences $\{a_i^{*,(k)}\}_{i=1}^{T-1}$ for $k = 1, \cdots, K$.

6      Update $\mu^{(j+1)}$ and $\Sigma^{(j+1)}$ as the mean and covariance of $\{a_i^{*,(k)}\}_{i=1}^{T-1}$, $k = 1, \cdots, K$.

7 **end**

8 Sample the action sequence $\{a_i^*\}_{i=0}^{T-1}$ from $\mathcal{N}(\mu^{(N)}, \Sigma^{(N)})$ as the output.

---

## D.2    LOW-LEVEL PLANNER

Note that the latent forward dynamics is given by $z_{t+1} = F(z_t, a_t)$. Without loss of generality, we consider $z_0$ as the current latent state and $z^*$ as the target latent state. Given a horizon $T \geq 1$, our low-level planner generates actions $\{a_t\}_{t=0}^{T-1}$ that drive the latent state $z_t$ to reach $z^*$ by solving a trajectory optimization problem

$$\underset{a_0,\cdots,a_{T-1}}{\text{minimize}} \quad \sum_{t=0}^{T} \|z_t - z^*\|^2 \tag{5}$$
$$\text{subject to} \quad z_{t+1} = F(z_t, a_t), \ t = 0, \cdots, T-1.$$

In this work, we apply CEM to solve the low-level planning problem (5). CEM has been successfully applied in model-based reinforcement learning (Finn & Levine, 2017; Wang & Ba, 2019; Hafner et al., 2019b) and is outlined in Algorithm Algorithm 3. In the CEM, the actions $\{a_i\}_{i=0}^{T-1}$ are drawn from a multivariate Gaussian distribution whose parameters (mean and covariance matrix) are updated iteratively to approximate the optimal distribution of actions.

