# OpenReview forum: "PcLast: Discovering Plannable Continuous Latent States"
_ICLR.cc/2024/Conference — Submitted to ICLR 2024_

### Official Review · Reviewer_wVZj · 2023-10-24

**Soundness:** 3 good
**Presentation:** 3 good
**Contribution:** 2 fair
**Rating:** 5
**Confidence:** 4

**Summary:**

This paper proposes a new way to learn "plannable" representations. Multi-step inverse dynamics models have the desirable property that the learned representations are guaranteed to contain only the "controllable" parts of states. However, they do not necessarily maintain a metric structure, and this lack of distance metrics makes it difficult to use for planning. To resolve this issue, the authors train a contrastive representation on top of the multi-step inverse dynamics-based representations to impose a metric structure. The authors show that this PCLast representations are indeed aware of the inherent temporal structure of the MDP, and that it leads to better performance in several types of environments.

**Strengths:**

- This work tackles the important problem of learning minimal representations that maintain sufficient information for control while having a metric structure. As far as I'm aware, this approach is the only such method (please correct me if not).
- The imposed metric structure enables hierarchical planning based on L2 distances on the representation space, and the authors show that this planning is indeed helpful for performance.
- The paper is well-written and easy to understand.

**Weaknesses:**

- The proposed method is highly complicated, involving five moving components: a forward dynamics model (FDM), a multi-step inverse dynamics model (IDM), a contrastive representation learner (CL), a hierarchical graph-based planner, and a low-level MPC planner, with several newly introduced hyperparameters for each module. While I see the motivation behind this approach (i.e., we want to first eliminate all the uncontrollable parts, and then impose a metric on another latent representation space), I'm not fully convinced that all of these components are necessary in practice (see the next bullet points for more specific concerns).
- In the Maze/Sawyer domains, the authors only compare PCLast with ACRO. Given the high complexity of the method, I think it is necessary to show that PCLast is at least better than all the "basic" representations that are used as building blocks for PCLast, i.e., IDM, CL, and FDM, to justify the added complexity.
- The performance of ACRO (the best baseline considered in this work) on the exogenous noise MuJoCo environments seems to be worse than the original performance reported in Islam et al. (2023) (e.g., in `cheetah_run_expert`). Could the authors explain this performance gap? Moreover, given the complexity of PCLast, I believe it requires more thorough empirical comparisons to demonstrate the effectiveness of PCLast (e.g., comparisons on the same set of environments (12 datasets x n distractor configurations) used in ACRO).
- Also, while the only "non-toy" benchmark considered in this work is MuJoCo in Section 4.5, it is unclear how planning helps on MuJoCo tasks (since hierarchical planners seem to be only used in the Maze environments). Given that the main benefit of PCLast is its ability to plan (as the title says), I believe it would be much more convincing if the authors could show this benefit in more practical environments.

[Minor]
- The paper is using the ICLR 2023 format.
- Typo (small $\\\|$) after "we formulate the cost as" in Appendix D.

I believe this is a borderline paper. While I appreciate the motivation behind PCLast, the empirical results are not strong/extensive enough to justify the need for all five components of PCLast.

**Questions:**

- What does LNS stand for in Appendix A?
- In Section 3.4, why do the authors use $\phi$ (the inverse dynamics model representation) instead of $\psi \circ \phi$ (the PCLAST representation)? Perhaps it is just a typo?

---

> ### Author Response · Authors · 2023-11-22
> **Response**
>
> > “... highly complicated, involving five moving components …”
>
> Yes, it may be possible to eliminate some components under certain conditions. But, in principle, each component of our method plays its own critical role for real-world scenarios i.e. multi-step inverse dynamics helps in learning exogenous noise free representation, forward dynamics helps with planning during inference , temporal contrastive representation helps with retaining metric space, hierarchical planner reduces plan computation time and robustifies it, and low-level MPC planner helps in navigating to a goal state. Please note, hierarchical planner and low-level MPC are non-learning components of our approach. They are required as we are in a reward-free setting and don’t learn any policy.
>
> In the future, we can consider potential approaches of merging these components into a single module. Also, if there is no exogenous noise, one can probably ignore inverse-dynamics.
>
> >  “… PCLast is at least better than all the basic representations… ”.
>
> In ACRO, authors have shown that their method works better than IDM and FDM. This is the reason that we focus on simply comparing it with ACRO and show limitations of ACRO.
>
> > “  The performance of ACRO (the best baseline considered in this work) on the exogenous noise MuJoCo environments seems to be worse than the original performance reported in Islam et al. (2023) (e.g., in cheetah_run_expert). Could the authors explain this performance gap? “
>
> In this paper, we evaluated on the hardest exogenous noise setting from the ACRO paper, which is called “HARD-EXO-Background Agent”, these are in Figure 20 of that paper’s appendix.  Our ACRO baseline is very similar to the numbers reported in that paper.
>
> >   “... how planning helps on MuJoCo tasks… ”
>
> When it comes to MuJoCo tasks, we simply focus on the method’s ability to learning a better representation than baselines. This empirically shows improvement in policy learning. This is also an example, where not all the components of our method are required to be used since the considered mujoco-tasks are not goal-conditioned.
>
> >   “...The paper is using the ICLR 2023 format…”
> We have fixed this.
>
> >   What does LNS stand for in Appendix A?
>
> LNS refers to “Local Neighbourhood Structure” which is enforced by the contrastive loss in Section 3.3. We have updated the text to be more clear about it.
>
> > “...authors use  (the inverse dynamics model representation) instead of  (the PCLAST representation)...”
>
> Yes, we use IDM representation in our work. Though, one can consider PcLast representation as well. The prime use of PcLast representation is to estimate distance between two states when planning.

---

> > ### Comment · Reviewer_wVZj · 2023-11-23
> >
> > Thanks for the response. I appreciate the roles of the five components, and agree that all of them are potentially useful in theory. However, I’m still not convinced that the newly added components, especially planning and contrastive learning (the main novelties of this work), play crucial roles _in practice_. I checked Figure 20 of the ACRO paper, but there still seems to be a significant gap between the original ACRO results and the results in this paper. Particularly, in `cheetah_run_medium_expert`, the original ACRO performance appears to be even better than that of PCLast. Moreover, even if PCLast outperforms ACRO in these settings, I still believe the empirical results are not comprehensive enough to justify the effectiveness of the new components given the small number (3) of "non-toy" tasks (in contrast, ACRO uses 84+). I maintain my rating this time.

---

### Official Review · Reviewer_uCK9 · 2023-10-30

**Soundness:** 2 fair
**Presentation:** 3 good
**Contribution:** 3 good
**Rating:** 3
**Confidence:** 3

**Summary:**

Latent representations of high-dimensional state spaces are a useful tool for
reinforcement learning, but existing approaches do not model structural
information about the environment. This paper proposes a new representation
learning approach, PCLaSt which ensures that states which are near each other in
the latent space have certain reachability properties. This is accomplished
using a secondary network to transform the latent space which is trained with a
contrastive loss. Enforcing reachability constraints on the state-space
representation is particularly useful in conjunction with hierarchical planning
algorithms, which rely on coherence between high- and low-level state-space
abstractions. Experimental results show that PCLaSt in combination with
hierarchical planning achieves much better performance than existing
latent-space learning approaches for goal-directed RL. PCLaSt also achieves
higher rewards in the presence of exogenous noise compared to baselines.

**Strengths:**

The algorithm is clean and well-presented, and the idea of using Brownian motion
to generate samples to learn reachability constraints is interesting.

The fact that PCLaSt can be used for both deep RL and more traditional path
planning (Table 1) demonstrates a good level of flexibility and suggests broad
potential impact.

The experiments which are presented show promising results, in particular
demonstrating better conformance between the latent space representation and the
true dynamics of the environment (Figure 4).

**Weaknesses:**

There are a several experiments for which results are not presented which I think
are necessary to clarify the benefits of this approach. In particular, the related
work lists HOMER and DRIML as methods which attempt to encode reachability
information in the latent space. However, these techniques are only included in
the experiments on exogenous information, and not in the experiments on
goal-conditioned RL. Similarly, I'd be interested in the results you can obtain
using the hierarchical planner with alternative latent space learning techniques
besides PCLaSt.

More generally, I'm a bit confused by the relation of PCLaSt to HOMER and DRIML.
The related work section argues that these two approaches are different from
PCLaSt because they don't deal with exogenous noise. However, in the technical
development of the paper, it seems that the denoising effects are due primarily
to ACRO, whereas the contribution of PCLaSt is primarily in enforcing
state-space geometry.

On a more minor notes, while the presentation is generally clear there is one
omission that I found confusing at first. In Section 3.3, the process for
generating samples for the contrastive loss omits a description of how $s_t$ and
$s_{t+k}$ are generated based on $y$.

**Questions:**

In general, reachability is not symmetrical (for example, it will not be
symmetrical in systems with momentum). However, the contrastive loss in Equation
2 minimizes the (symmetrical) distance between states when one is reachable from
the other. Is there a theoretical or intuitive reason that this mismatch
wouldn't hinder planning in systems with highly non-symmetrical reachability
properties?

From my understanding of the paper, it seems that the noise filtering effects of
PCLaSt are largely shared with ACRO. Is there some explanation of why PCLaSt
seems to be so much more effective in Figure 6?

---

> ### Author Response · Authors · 2023-11-22
> **Response**
>
> >  “… HOMER and DRIML …”
> We primarily compete and build upon ACRO as it showed that it does better than HOMER and DRIML to remove exogenous noise.  Note that HOMER uses a leveled structure which is unwieldy in practice (it’s a theoretical algorithm) and DRIML uses rewards so it requires a different information access structure than PCLast to apply.
>
> > “… Hierarchical Planner with other latent state spaces…”
> As we investigated with other latent-state representations, we found that they lead to poor clustering of states for abstraction as shown for ACRO in Fig 1 (b) and Fig 4 ( third-row); making them unsuitable for further investigation.
>
> > In general, reachability is not symmetrical (for example, it will not be symmetrical in systems with momentum). However, the contrastive loss in Equation 2 minimizes the (symmetrical) distance between states when one is reachable from the other. Is there a theoretical or intuitive reason that this mismatch wouldn't hinder planning in systems with highly non-symmetrical reachability properties?
>
> This is an interesting aspect, and we can consider investigating it further in future work. One probable approach would be extend our current loss with weighted loss based on some reachability score of states in a trajectory.
>
> > “…PCLaSt seems to be so much more effective in Figure 6?”
> PCLast helps in learning representation which maintains metric space between states. Intuitively, this eases the inter and extrapolation between states for policy, helping with better generalization.

---

### Official Review · Reviewer_Gktz · 2023-11-01

**Soundness:** 3 good
**Presentation:** 3 good
**Contribution:** 3 good
**Rating:** 5
**Confidence:** 4

**Summary:**

This paper proposes a method for learning a hybrid continuous-discrete representation that supports hierarchical planning. The first component of the method is an image encoder that learns a latent space using an inverse modeling loss. The second component is an additional encoder that re-projects the latent space so that it can be clustered using k-means; this additional encoder is trained using time-contrastive learning. The third component is a forward model learned in the latent space. Finally, a hierarchical planner finds the shortest path through state clusters and a low-level planner takes care of transitioning between clusters. The system is tested in various maze-like and continuous control environments. Experiments include varying the number of levels in the hierarchy.

**Strengths:**

I reviewed a previous version of this paper.

1. I appreciate the work that went into improving this paper since its last version. The paper now contains extensive experiments including a diverse set of domains and several different learning scenarios.

2. The paper proposes a very interesting combination of multiple latent spaces and losses to support hierarchical planning. I find this to be an important problem in RL and robotics.

3. The method is demonstrated in online and offline reinforcement learning in several different environments.

**Weaknesses:**

1. The first main contribution is a new contrastive loss function. Although I like the justification of the loss function through the model of Brownian motion (Appendix B), there are several previously proposed loss functions that seem to solve the same problem. Specifically, the two time-contrastive loss functions I list below pull embeddings of frames nearby in time together and push embeddings of frames far away in time or from completely different videos apart. In the related work, the authors state that time-contrastive learning “is not invariant to exogenous information”, but neither is the proposed loss function. Invariance to exogenous information is achieved by first training the latent space using a prior method, ACRO.

The loss function proposed in this paper:
$- \log \sigma ( \beta - \alpha D(s_t, s_{t+k}) ) - \log (1 - \sigma ( \beta - \alpha D(s_t, s_r) ) )$

Wang et al., 2015:
$max(0, D(s_t, s_{t+k}) - D(s_t, s_r) + m)$

Nair et al., 2022:
$- \log \frac{\exp S(s_t, s_{t+k}) }{\exp S(s_t, s_{t+k}) + \exp S(s_t, s_{t+k+l}) + \exp S (s_t, s_r)}$

Where $D$ is a distance function and $S$ is a similarity function between a pair of states.

2. The second main contribution is the entire system of first learning an encoder $\phi$ using an inverse dynamics loss function, followed by a contastively-learned encoder $\psi$, followed by a forward model, followed by K-means clustering, followed by a hierarchical planner. I would like to see more justification of this approach. For example, the combination of $\psi(\phi(x))$, where $\phi$, is trained first and frozen, is not fully justified. Could we train a single encoder with multiple losses? Further, the choice of the hierarchical clustering and planning methods could be compared to other hierarchical systems (e.g. Kurutach et al., 2018).

3. (Relatively minor) From the introduction to the methods section, the paper does not flow very well. I think the reason is that the paper lacks focus on its key contribution. If the key contribution was a new and well-justified contrastive learning function, the paper could be written differently. If the key contribution was a system of learning a hierarchical latent space, the paper could also be written differently. This might be just a matter of personal taste, so feel free to disregard.

Minor:

* Should “τ ≫ t ≫ t0” be “τ >= t >= t0”?
* As I stated in my previous review, the proof of Proposition 1 contains the term $P_k(z′|z′, y = 1)$, which I think is a typo since $z’$ appears twice?

References:

Wang et al., 2015: https://arxiv.org/abs/1505.00687

Nair et al., 2022: https://arxiv.org/abs/2203.12601

Kurutach et al., 2018: https://arxiv.org/abs/1807.09341

**Post rebuttal:** Thank you for answering my questions! Unfortunately, I do not think this paper can be accepted without more comparisons to prior time-contrastive learning losses as well as prior hierarchical learning methods. I highlighted these in my review. Nevertheless, I believe the work addresses an important problem and is eventually going to be a strong submission.

**Questions:**

1. How does your proposed loss function compare to the previously proposed time-contrastive learning losses?

2. Is it possible to train a single state encoder using both the inverse dynamics and contrastive learning losses?

---

> ### Author Response · Authors · 2023-11-22
> **Response**
>
> > How does your proposed loss function compare to the previously proposed time-contrastive learning losses?
>
> On Wang et. al, the use of a margin is likely to be sufficient for using the representations downstream for classification, but isn't likely to work for capturing the precise numerical details in state position that are needed for control. On the Nair 2022 loss, likewise, a similarity won't necessarily yield a correct local neighborhood structure.
> Nonetheless these are closely related and citations have been added, thanks for pointing these out.
>
> > Is it possible to train a single state encoder using both the inverse dynamics and contrastive learning losses?
>
> There are known counterexamples where mixing contrastive temporal and multi-step inverse leads to contamination of the latent state with exogenous noise.  Even a small weighting on contrastive temporal in a mixed loss create this contamination effect.

---

### Official Review · Reviewer_AtR8 · 2023-11-07

**Soundness:** 2 fair
**Presentation:** 3 good
**Contribution:** 2 fair
**Rating:** 5
**Confidence:** 3

**Summary:**

This paper proposes a state representation learning approach, PCLaSt, which associates state reachability so that nearby and reachable states can be clustered together in the latent space. This learned representation can be directly used for online and offline RL, especially for goal-reaching tasks. Moreover, with this reachability-aware latent representation, we can efficiently find a path from one state to another by clustering the latent states, forming a graph of the clusters with their connectivity (reachability), and running Dijkstra's algorithm on the graph. The experiments on 2D maze navigation and 3D Sawyer reaching show that the proposed representations improve goal-reaching task performances in diverse RL settings.

**Strengths:**

* The paper is very well written. The motivation and the proposed approach are easy to understand.

* The idea of embedding reachability into state representations is intuitive and its implementation with contrastive learning is straightforward.

* Visualization in Figure 1 and Figure 4 is very helpful in understanding how useful the latent states PCLaSt could learn.

**Weaknesses:**

* Throughout the paper, there are several symbols used to denote different levels of latent states. However, each of the symbols $x$, $z$, and $s$ sometimes means different levels of abstraction. It might be easier to follow if each symbol is used to represent a single entity and a summary of these symbols is illustrated as a figure or list. If I didn't understand correctly, the paper writing could be improved to make it straightforward.

* The proposed approach makes sense in 2D/3D navigation domains where reachability can be represented with a very low dimensional (2D/3D) latent space. However, it is unclear whether this also applies to more complicated environments with high-dimensional C-space, e.g., robotic manipulation with many objects. If the proposed method is not scalable but still useful for navigation tasks, the paper needs to explain which family of problems can be handled by the proposed approach and which family of problems cannot.

* The proposed planner assumes that latent states in the same cluster are reachable from each other, which may not be true since the latent state representations are approximated and learned from random exploration trajectories. It might not be a problem in the experiments in this paper since the experiments are done in simple 2D navigation and 3D reaching tasks. However, this may fail when a higher-level plan cannot be achieved because an agent cannot reach a waypoint (the center of a cluster in the plan) from the current state. It is required to discuss this issue and how to resolve it.

* The results in Sawyer-Reach do not demonstrate the advantage of the proposed approach as explained in Section 4.3. Extensive experiments with more challenging environments that can show the benefit of the proposed approach would strengthen the paper.

* The experiments rely on exploratory data to pre-train the representations. This may not be scalable to more complex environments with many narrow passages in C-space. One alternative could be learning the representations using play data or offline data. This can be an interesting experiment to see.

**Questions:**

Please address the weaknesses mentioned above.


### Minor questions and suggestions

* In Section 3, $x$ is often used for a variable in both $\phi(x)$ and $\psi(x)$. But, since $x$ is used to describe an observation in the section, it might be better denoted as $\psi(s)$. Similarly in Section 3.2, $z$ is introduced for a latent state but it could be simply $s$?

* In Section 3.2, $f_\text{AC}$ is defined to take two latent states but the following sentence says it takes a concatenation of $\tau + 1$ latent states, which is confusing.

* In Section 3.3, it could be clearer if the latent state spaces induced by $\phi$ and $\psi$ are defined separately, like $\mathcal{S}$ and $\mathcal{Z}$? The definition of the second encoder $\psi(x): \mathcal{Z} \rightarrow \mathcal{Z}$ is kinda confusing as these two spaces are not shared (although they can have the same dimensionality).

* Figure 2 is useful to understand the holistic view of the proposed method. However, it is a bit overwhelming with many symbols. It could be easier to understand this figure if it includes a few image observations for $x$s and uses different shapes or colors for different symbols.

---

> ### Author Response · Authors · 2023-11-22
> **Response**
>
> Thanks for taking time to review our work. In the following, we address each of your concerns:
>
> > ...there are several symbols used to denote different levels of latent state…
>
> With “x”, we simply imply observations from the environment; “z” refers to the learnt latent state from the multi-step inverse-dynamics model, and “s” refers to the true-latent state. Though we attempted to be consistent in our usage of these symbols, we will revise our document to fix any inconsistencies and re-emphasize them in our “Section 3.1 Notation and PRELIMINARIES”
>
> > ...more complicated environments with high-dimensional C-space…
>
> In principle, our work is not restricted to the dimensionality of the true state. We learn our latent state representation from high-dimensional image inputs and keep the dimensionality of the latent state in the order of 128 or 256 for our experiments. Our work rather emphasizes learning latent states for any high-dimensional environment which requires goal conditioned planning. Our experiments on half-cheetah and walker-2d suggest our learning method seems to be helping.
>
> > ... The proposed planner assumes that latent states in the same cluster are reachable from each other …
>
> Thanks for pointing out our assumption. Yes, we do make this assumption and it’s based on learning formulation where latent states are learned in a manner such that state which are reachable to each other have similar latent-states. This helps in ensuring that any state which is clustered-together is navigable; irrespective of “k” used for k-means clustering. This is subject to learning latent state representation up to convergence.
>
>
> > ... One alternative could be learning the representations using play data or offline data …
>
> Thanks for the suggestion. In our current-experiments with half-cheetah and walker 2d, we make use of offline  “medium, medium-expert, and expert” datasets.
>
> > ... it might be better denoted as \psi(s)...
>
> We have made the suggested change in the later half of section 3.1. For section 3.2, we continue to use “z” to maintain continuity with the rest of the document.
>
> > ... defined to take two latent states but the following sentence …
>
> Yes, it takes concatenation of the z_t, z_{t+k} and embedding of “k”. We have reworded this in section 3.2 to make it more clear. Thanks for pointing it out.
>
> >  ...the definition of second encoder...
>
> We have updated the definition in section 3.3
>
> > ...However, it is a bit overwhelming with many symbols…
>
> Thanks for the suggestion, we will update the figure to have different symbols and colors for different modules.

---

> > ### Comment · Reviewer_AtR8 · 2023-11-23
> >
> > Thank for the author response.
> >
> > The reachability assumption inside a cluster is not convincing to me, especially when the true state space becomes high-dimensional. Strong empirical results on tasks with high-dimensional state spaces would be required to claim the effectiveness of the proposed approach beyond simple 2D/3D navigation environments. Thus, I would keep my original rating of weak rejection.

---

### Meta-Review · Area_Chair_hAiH · 2023-12-05

**Metareview:**

This paper introduces PCLaSt, a method to learn a concept of reachability such that reachable states can be clustered together in the latent space, which makes hierarchical planning much easier. PCLaSt has many components to allow for the pipeline where it learns a latent representation, clusters such latent states, learns a forward model, and then performs hierarchical planning. Overall, the reviewers agreed that the paper introduces interesting ideas and that it is well written. However, concerns have been raised about (1) the generality of assumptions the paper makes, (2) the lack of comparison to other baselines, and (3) some uncertainty about the overall performance when compared to other papers. The responses provided in the discussion phase were short and some seemed even incomplete (e.g., reviewer uCK9), and they were judged to be unsatisfactory. This is not a decision I take lightly, but I’m recommending the rejection of this paper. I believe the paper could be improved for a future submission if it addressed some of the concerns raised by the reviewers.

**Justification For Why Not Higher Score:**

All reviewers agree with this paper being rejected. As I wrote in the meta-review, concerns have been raised about (1) the generality of assumptions the paper makes, (2) the lack of comparison to other baselines, and (3) some uncertainty about the overall performance when compared to other papers. The responses provided in the discussion phase were short and some seemed even incomplete (e.g., reviewer uCK9), and they were judged to be unsatisfactory.

**Justification For Why Not Lower Score:**

N/A

---

### Decision · Program_Chairs · 2024-01-16

Reject